# Evaluation of the Antioxidant Activities and Phenolic Profile of Shennongjia *Apis cerana* Honey through a Comparison with *Apis mellifera* Honey in China

**DOI:** 10.3390/molecules28073270

**Published:** 2023-04-06

**Authors:** Jingwen Guo, Qiong Ding, Zhiwei Zhang, Ying Zhang, Jianshe He, Zong Yang, Ping Zhou, Xiaoyan Gong

**Affiliations:** 1College of Chemistry and Molecular Sciences, Wuhan University, Wuhan 430072, China; 2AB Sciex Co., Ltd., Beijing 100102, China

**Keywords:** *Apis cerana* honey, *Apis mellifera* honey, antioxidant capacity, polyphenols profile, metabolomics

## Abstract

This study evaluates the phenolic profile as well as the antioxidant properties of Shennongjia *Apis cerana* honey through a comparison with *Apis mellifera* honey in China. The total phenolic content (TPC) ranges from 263 ± 2 to 681 ± 36 mg gallic acid/kg. The total flavonoids content (TFC) ranges from 35.9 ± 0.4 to 102.2 ± 0.8 mg epicatechin/kg. The correlations between TPC or TFC and the antioxidant results (FRAP, DPPH, and ABTS) were found to be statistically significant (*p* < 0.01). Furthermore, the phenolic compounds are quantified and qualified by high performance liquid chromatography-high resolution mass spectrometry (HPLC-HRMS), and a total of 83 phenolic compounds were tentatively identified in this study. A metabolomics analysis based on the 83 polyphenols was carried out and subjected to principal component analysis and orthogonal partial least squares-discriminant analysis. The results showed that it was possible to distinguish *Apis cerana* honey from *Apis mellifera* honey based on the phenolic profile.

## 1. Introduction

Honey can serve as a source of natural antioxidants. The antioxidant activity of honey is primarily provided by its polyphenols [1]. Thus, a considerable variation of antioxidant activity and polyphenols profile is found among different honey varieties around the world [2,3,4,5]. This variation is mainly due to different floral and geographical origins as well as the type of bees [6,7]. Therefore, the analysis of phenolic profile has been regarded as a very promising technique for studying the floral, geographical and honeybee origins of honeys.

In general, *Apis cerana* (*A. cerana*) honey is produced by *Apis cerana* grazing on various botanical sources. Traditionally, *A. cerana* honey is more nutritious than other honey varieties because of its long nectar cycle and the wide variety of nectar source [8]. There are recent research reports that have found various benefits for *A. cerana* honey, such as its’ anti-inflammatory, anti-oxidant, and beneficial effects with regard to acute alcohol-induced liver damage [9,10,11]. These therapeutic activities have been attributed to the phenolic acid and flavonoids content of *A. cerana* honey [9,11]. Nonetheless, there is still a lack of understanding about the phenolic profile and antioxidant capacity of *A. cerana* honey. Until now, most of the studies have focused mainly on the phenolic profile and antioxidant activity in mono-floral honeys from *Apis mellifera* (*A. mellifera*) in China [12,13,14,15,16,17]. Therefore, it is necessary to evaluate the phenolic compounds in *A. cerana* honey here.

In ancient China, “Shen Nong’s Herbal Classic” have recorded the use of *A. cerana* honey from Shennongjia district as the first use of medicine. The Shennongjia forestry district is the only well-preserved subtropical forest ecosystem in the middle latitudes of the world, located in Hubei province in China. Due to its superior climatic conditions and unique geographical environment, the pollen of abundant resources of bee plants and wild medicinal plants is a good source of honey, and the honey is the “shennong poly-floral honey” with local characteristics. As far as we know, there are no reports about the phenolic profile of Shennongjia *A. cerana* honey. Thus, the antioxidant activity and polyphenol profile of this honey are analyzed in this study. Moreover, considering that Shennongjia *A. cerana* honey is poly-floral honey; hence, mono- and poly-floral honey from *A. mellifera* are selected to systematically evaluate the polyphenol profile between the two honey groups.

Furthermore, the recovery of phenolic compounds from honey varies differently depending on the pre-concentration methods [15,18,19]. Thus, to reduce metabolite information losses due to the different extraction methods, phenolic compounds in honeys were isolated using liquid-liquid extraction and solid-phase extraction (SPE) methods, respectively. Afterwards, based on the profile of phenolic compounds obtained by various extraction methods, a metabolomics analysis was carried out and the honey was subjected to a principal component analysis and an orthogonal partial least squares-discriminant analysis. The secondary aim of the present study was to differentiate *A. cerana* honey and *A. mellifera* using multivariate techniques.

## 2. Results and Discussion

### 2.1. Total Phenolic Content (TPC), Flavonoid Content (TFC), and Antioxidant Activity

The TPC, TFC, and antioxidant activity levels of *A. cerana* (A.c_1 to A.c_26, *n* = 26) and *A. mellifera* (A.m_p1 to p8 from Wuhan, *n* = 8; A.m_F from Fangxian) honeys from China and two Manuka honeys from New Zealand were evaluated, and the results are presented in Table 1.

The values for TPC in *A. mellifera* and *A. cerana* honeys in China ranged from 104.33 ± 4.21 to 379.20 ± 25.86 mg GAE/kg and from 263.02 ± 2.23 to 680.90 ± 35.80 mg GAE/kg, respectively. The values for TFC in *A. mellifera* and *A. cerana* honeys from China ranged from 14.74 ± 0.71 to 42.76 ± 0.29 mg EC/kg, and from 35.87 ± 0.44 to 102.24 ± 0.75 mg EC/kg, respectively. Here, A.c_8 and A.c_6, two *A. cerana* honeys from the Shennongjia region, had the highest TPC and TFC values, respectively, while the lowest TPC and TFC were measured in A.m_p1 and A.m_p7 honey, respectively. The range of values for the TPC and TFC here reported were in agreement with those previously found in Chinese honeys from *A. cerana* and *A. mellifera* [9,20]. In addition, previous reports showed similar TPC and TFC amounts for mono-floral and poly-floral honeys from other geographical origins [2,6,21,22].

Table 1 also showed the FRAP, DPPH and ABTS values for different honey samples in China. Two *A. cerana* honeys, including A.c_1 and A.c_8, had the highest FRAP (A.c_1), DPPH (A.c_1), and ABTS (A.c_8) values. Interestingly, A.m_p1 and A.m_p7 honeys had the lowest levels of TPC and TFC, and, correspondingly, the lowest values of FRAP, DPPH, and ABTS. Moreover, the correlation analysis results showed that there was a correlation between TPC or TFC and the levels of FRAP, DPPH, and ABTS (*p* < 0.01), suggesting that phenolic compounds are some of the main species responsible for the antioxidant capacity of honey [2]. The correlation between TPC or TFC and the levels of antioxidant activity here reported was in agreement with the results previously reported by other authors [23,24].

Furthermore, as a control, two Manuka honeys from New Zealand had higher TPC, TFC, and antioxidant activity levels than most of the *A. cerana* and *A. mellifera* honeys in China in this study. This means that in addition to the influence of bee species, plant and geographical sources can also affect the content of polyphenols [6,7].

### 2.2. Quantification of Thirteen Polyphenols in Honeys Using Different Extraction Methods

Several common phenolic compounds and abscisic acid in honey that are reported in the literature were isolated using three different extraction methods and quantified in the present study. The LOD (Limit of Detection), LOQ (Limit of Quantitation), linear range, and MS characteristics of these compounds are listed in Appendix A. 

Table 2 shows the average amount of each compound isolated using different methods in the *A. cerana* and *A. mellifera* honeys. As seen, the average content of thirteen polyphenols in samples varied considerably depending on the extraction methods. EAC (ethyl acetate, liquid-liquid extraction) generated higher levels of kampferol (*p* < 0.0001), quercetin (*p* < 0.0001), vanillic acid (*p* < 0.0001), and trans-ferulic acid (*p* < 0.01), while, SPE (XA and PLS, solid-phase extraction) generated higher levels of rutin (*p* < 0.0001). Between the two SPE cartridges, the Strata XA cartridge showed lower recoveries of vanillic acid (*p* < 0.0001) and 4-hydroxybenzoic acid (*p* < 0.0001) compared to ProElut PLS SPE cartridges. The results suggested that different extraction methods have different extraction efficiency for phenolic acids and flavonoids.

In general, most of the flavonoids showed a lower average content than phenolic acids. Among thirteen compounds, kaempferol and 4-hydroxybenzoic acid were the main flavonoid and phenolic acid found in the *A. mellifera* and *A. cerana* honeys in China. It was reported that kaempferol and 4-hydroxybenzoic acid were prevalent in *A. mellifera* honey of different geographic origins in previous studies [14,25,26,27]. In addition, chrysin was present at the lowest levels in *A. cerana* honey, while rutin had the lowest content in *A. mellifera* honey. Furthermore, some flavonoids showed significant distinctions between *A. cerana* and *A. mellifera* honeys regardless of extraction methods. For example, the contents of quercetin (*p* < 0.05), rutin (*p* < 0.01) and p-coumaic acid (*p* < 0.05) were higher in *A. cerana* honeys than those in *A. mellifera* honeys. Three compounds including pinocembrin (*p* < 0.01), chrysin (*p* < 0.01), and galangin (*p* < 0.001) were considered as propolis-derived flavonoids [22,25], had lower contents in *A. cerana* honeys.

### 2.3. Identification of Individual Polyphenols

One hundred and eleven honey extracts were subjected to the identification of the flavonoids, phenolic acids and abscisic acid based on the optimization conditions of HPLC-QTOF-MS/MS. A total of 83 compounds were tentatively identified, and 13 of them were qualified by comparing the retention times (RT) and the MS spectra with available standards. In the absence of standards, the identification of a further 70 compounds was based on the search for the [M–H]^−^ deprotonated molecule and its fragmentation referred to in the literature. Table 3 summarizes the data obtained for each of the identified compounds with their retention times, error in ppm (between the mass found and the accurate mass), as well as the MS/MS fragment ions.

Hydroxycinnamic acids such as caffeic acid and their derivatives were the main phenolic acids found in the study. Caffeic acid was present in all of the honey samples; in addition, ten caffeic acid derivatives were detected: caffeoylquinic acid isomers (compounds 6, 14 and 17), dicaffeoylquinic acid isomers (compounds 25, 27 and 30), and four ester derivatives of caffeic acid (compound 31, 32, 33 and 34). All of the caffeic acid derivatives showed negative product ions at 179 m/z due to the loss of the deprotonated molecule of caffeic acid. Caffeoylquinic acids and dicaffeoylquinic acids were reported in the European honeydew honey [33] and *A. mellifera* honey from different botanical and geographical origins [2,25,27,28,37]. Caffeic acid ester derivatives were detected in Chilean propolis [30] and Spanish *A. mellifera* honey [2]. As shown in Table 3, caffeic acid ester derivatives were commonly present in *A. mellifera* honey in this study, whereas they were relatively rare in *A. cerana* honey.

Furthermore, both isomers of abscisic acid previously described in other varieties of honey [2] were detected in all of the honey samples in the study. In addition, 4-ethoxy-3-methoxycinnamic acid (compound 24) was identified only in Manuka honeys in the study. By examining the empirical formula of this compound, it was concluded that it may be an ethylated derivative of ferulic acid. It produced MS2 fragments at 193, 179, 151, and 135 m/z, most probably corresponding to [M−H−C_2_H_5_]^−^, [M−H−C_2_H_5_−CH_3_]^−^, [M−H-C_3_H_3_O_2_]^−^ and [M−H-C_3_H_3_O_2_−CH_3_]^−^ fragments, respectively.

Concerning flavonoids, four subclasses of compounds were identified: flavonols, flavanonols, flavanones, and flavones, in addition to some flavanonol ester derivatives and flavonols glycosides. The flavanonol ester derivatives mainly came from pinobanksin (compounds 58, 59, 60 and 61), which showed a negative product ion at 271 *m*/*z* due to the loss of the deprotonated molecule of pinobanksin. Pinobanksin and its ester derivatives are characteristic flavonoids of propolis, and were found in Spanish *A. mellifera* honeys [2], sulla honey from the Sicilian native breed of black honeybee [36], as well as the Chilean propolis [30]. In this study, these compounds were present in almost all *A. mellifera* honeys, but very few were found in *A. cerana* honey. For example, pinobanksin-3-O-hexanoate (compound 61) was present in all *A. mellifera* honeys except for A.m_p7 honey, while it was undetectable in all *A. cerana* honey samples (Table 3).

The flavonols’ glycosides that were mainly from quercetin, kaempferol, methoxykaempferol, and isorhamnetin were previously described in different types of honey [2,33,34]. Numerous derivatives of flavonols’ glycosides were identified in *A. mellifera* and *A. cerana* honey extracts in this study: rhamnosides (loss of 146 Da), hexosides (loss of 162 Da), neohesperidoside, rhamnosylhexoside (loss of 308 Da), and dihexosides (loss of 324 Da). For example, in MS2 spectra of compound 47 at 46.92 min and 431 *m*/*z*, base peak fragments at 285 *m*/*z* (loss of 146 Da) and additional two fragment ions resulting from the loss of 257 and 151 Da could be observed, and it was then concluded that it could be kaempferol–rhamnosides.

In conclusion, propolis-derived caffeic acid and pinobanksin ester derivatives were widely present in *A. mellifera* honeys in the study, but rarely in *A. cerana* honeys.

### 2.4. Metabolomics Analysis

A PCA was conducted to evaluate the effect of the honey species / extraction method on the 83 phenolic compounds from a descriptive point of view (Figure 1). As shown in a PCA scores plot (Figure 1A), all of the *A. cerana* honey extracts regardless of extraction method (*n* = 77) were designed in PC1 negative, and most of the *A. mellifera* honey extracts (*n* = 29) were designed in PC1 positive. These results suggested that different honey species, rather than extraction methods, could be distinguished based on the levels or the presence of phenolic compounds.

For *A. cerana* honeys distributed in PC1 negative, most of the honey extracts (*n* = 72) clustered tightly, except for five honey extracts which were far away from other honey extracts due to their high level of methoxy kaempferol (Figure 1B). For *A. mellifera* honeys distributed in PC1 positive, the poly-floral *A. mellifera* honey extracts (*n* = 24) clustered tightly and were closest to the *A. cerana* honey group, followed by mono-floral *A. mellifera* (A.m_F) honey, and then by Manuka honey. The result indicated that botanical and geographical origins have an effect on the phenolic profile in *A. mellifera* honeys. Manuka honey was differentiated from other honeys for the high contents of 4-methoxyphenyllactic acid and p-hydroxy-hydrocinnamic acid. Fangxian *A. mellifera* honey was monofloral honey and characterized by a high content of pinobanksin (Figure 2B). Wuhan *A. mellifera* honey was polyfloral honey, and thus it may be closer to Shennongjia *A. cerana* honey in its phenolic acid profile because of the diversity of plant sources.

Furthermore, an orthogonal partial least squares-discriminant analysis (OPLS-DA) was conducted to analyze the differences between *A. mellifera* and *A. cerana* honey. Figure 2A showed that *A. cerana* honey samples were located on the right side of the ellipse and were well separated from *A. mellifera* honey samples. This result indicated that there were significant differences in the two honey groups. In addition, seven-fold cross-validation and 200 permutations were conducted to further verify the predictability of the OPLS-DA model. As shown in Figure 2B, the intercept of Q^2^ (−0.223) was negative on the vertical axis, and all blue Q^2^-values to the left were lower than the original points to the right, indicating that the established model was not overfitted for the experiment.

The variables responsible for discriminating *A. cerana* from *A. mellifera* honey were then identified using the OPLS-DA VIP (Figure 2C, VIP > 1) and S-plot (Figure 2D). The red variables (Figure 2C, VIP > 1) were tested using a Student’s t-test and the corresponding VIP and *p* values (*p* < 0.01) are listed in Appendix A. An S-plot (Figure 2D) was used to visualize the covariance and correlation between *A. mellifera* and *A. cerana* honey. Here, eight variables (compound 1–8 in Appendix A, *p* < 0.01) were far from the origin and were located at the far left of the X-axis. This indicated that the contents of these compounds in *A. mellifera* honey were higher than those in *A. cerana* honey. Among these compounds, five propolis-derived flavonoids (pinobanksin, pinobanksin-5-methyl ether, galangin, chrysin and pinocembrin), were commonly present in all *A. mellifera* honeys in the present study (Table 3). These flavonoids have previously been identified in propolis, European honeydew honey, and mono- and polyfloral honey from *A. mellifera* [2,27,30,33,38].

As shown in Figure 2D, five variables (compound 9–13 in Appendix A, *p* < 0.01) were far from the origin and were located at the far right of the X-axis. The result indicated that the contents of these compounds in *A. cerana* honey were higher than those in *A. mellifera* honey. The five compounds have been previously reported in tilia, salvia officinalis L., and chestnut source honey samples [2,25,39]. In this study, they were commonly present in *A. cerana* and *A. mellifera* honey. The high content levels of these compounds in Shennongjia *A. cerana* honey may be due to the abundant sources of wild medicinal plants and nectar plants in this region.

Of course, whether these compounds can be used as appropriate markers to distinguish *A. cerana* honey from *A. mellifera* honey requires further study and confirmation by expanding the sample size and selecting *A. cerana* and *A. mellifera* honey from different geographical and plant sources in the future.

## 3. Materials and Methods

### 3.1. Chemicals

All solvents and phenolic compounds used for HPLC analysis were of HPLC grade, and the rest of the chemicals were of analytical grade. Phenolic compounds including caffeic acid (Cafa), trans-cinnamic acid (Tcina), chrysin (Ch), trans-ferulic acid (Fera), galangin (Gal), *p*-coumaric acid (Pcoa), vanillic acid (Vana), 2-cis-4-trans-abscisic acid (CTabsa), kaempferol (Kaem), 4-hydroxybenzoic acid (4Hba), and quercetin (Quer) were obtained from Sigma–Aldrich. Rutin (Ru), pinocembrin (Pino), gallic acid and epicatechin were from the Bei Na Chuang Lian Institute of Biotechnology. Folin Ciocalteu’s phenol reagent, 1,1-diphenyl-2-picrylhydrazyl (DPPH), 2,4,6-tri(2-pyridyl)-1,3,5-triazine (TPTZ), and 2,2′-azino-bis (3-ethylbenzothiazoline-6-sulphonic acid) diammonium salt (ABTS) were purchased from the Sinopharm Chemical Reagent Co., Ltd (Shanghai, China). Solid-phase extraction cartridges Strata-X-A (60 mg/3 mL) and ProElut PLS (60 mg/3 mL) were acquired from Phenomenex Inc. (Torrance, CA, USA) and Dikma Technologies Inc. (Beijing, China), respectively.

### 3.2. Honey Samples

The poly-floral honeys were harvested from *A. cerana* (Shennongjia; *n* = 26, 110°40′ E, 31°44′ N) between September and October 2017. The poly-floral honeys from *A. mellifera* (Wuhan; *n* = 8; 114°21′ E, 30°28′ N) were purchased from the Kangsinong Bee technology Co. Ltd. (Wuhan, China) in 2017. July mountain flower honey from *A. mellifera* (A.m_F) in Fangxian (110°44′ E, 32°3′ N) was collected in Hubei province. Two Manuka honeys (MGO100+ and MGO250+) from New Zealand, as controls, were purchased from Amazon.com, Inc. (Seattle, WA, USA) in 2017. The floral origins of *A.cerana* and *A. mellifera* honey samples were determined by the melissopalynological analysis, as previously reported [40]. The results are listed in Appendix A.

### 3.3. Extraction of Phenolic Compounds

The extraction of phenolic compounds was undertaken by solid-phase extraction. The SPE method was carried out according to the previous study [15] with minor modifications. A total of 10.0 g of honey samples were mixed with 50 mL of ultrapure water, and then the solution was adjusted to pH = 2 with HCl for the PLS cartridges or adjusted to pH = 7 with 5% ammonium (*v*/*v*) for the Strata X-A cartridges. After removing the impurity particles by centrifugation (8000 g, 10 min), the supernatants were loaded onto the previously conditioned cartridges (according to the manufacturer’s instructions). After loading, these cartridges were washed with 4 mL of acidified ultrapure water (pH = 2) for the PLS SPE cartridges or washed with 4 mL of ultrapure water (pH = 7) for the Strata X-A SPE cartridges. The phenolic compounds retained on the cartridges were then eluted with 5 mL of formic acid: methanol (1:9, *v*/*v*). The extract was evaporated at 40 °C under a stream of nitrogen, and then reconstituted in 1 mL of methanol with 0.1% formic acid. The obtained extracts were filtered and stored at −20 °C until further analysis by high performance liquid chromatography–quadrupole time-of-flight mass spectrometry (HPLC–QTOF-MS).

The extraction of phenolic compounds was undertaken with liquid-liquid extraction. Briefly, 10.0 g of honey samples were mixed with 50 mL of ultrapure water, and the solution was then adjusted to pH = 2 with HCl. The honey solution was extracted three times with 20 mL of ethyl acetate. The extracts were evaporated to dryness on a rotary evaporator at 30–40 °C, and then dissolved in 1 mL of methanol with 0.1% formic acid. The obtained extracts were filtered and stored at −20 °C until further analysis by HPLC–QTOF-MS.

### 3.4. HPLC–QTOF-MS Conditions

HPLC analyses were performed using a Shimadzu LC-20A system (Shimadzu Corporation, Kyoto, Japan) coupled with a quadrupole time-of-flight mass spectrometer (AB Sciex Triple QTOF5600+, AB Sciex, Redwood, CA, USA). The chromatographic separation was carried out using an Eclipse XDB-C18 column (100 mm × 2.1 mm, 3.5 um) (Agilent, Wilmington, DE, USA). The mobile phase consisted of 0.1% formic acid in water (phase A) and 0.1% formic acid in methanol (phase B). The flow rate was 0.3 mL/min and the injection volume was 10 µL, while the temperature of the column oven was set at 35 °C. The gradient separation was performed as follows: 0–1 min, 0% (B); 1–6 min, 0–6% (B); 6–13 min, 6–10% (B); 13–25 min, 10–20% (B); 25–35 min, 20–40% (B); 35–40 min, 40% (B); 40–55 min, 40–65% (B); 55–60 min, 65% (B); 60–75 min, 65–98% (B); 75–80 min, 98% (B). TOF–MS and the data of ten TOF–MS/MS were collected in negative ion mode using the information-dependent acquisition (IDA) function. The parameters were as follows: dynamic background subtraction (DBS); charge monitoring to exclude multiply charged ions and isotopes; Ion Source Gas1: 55 psi; Ion Source Gas2: 60 psi; Curtain Gas: 30 psi; Temperature: 600 °C; IonSpray Voltage Floating: −4500 V; Declustering Potential: 100 V; Collision Energy: 25 V; Collision Energy Spread: 15 V. In order to ensure the stability of outcomes, the calibration reagent (sodium formate) was detected in every two sample intervals, and methanol was used as a blank control to avoid the misjudgment of characteristic markers. In parallel, quality control (QC) samples were prepared by mixing equal volumes (9 µL) from each sample. An aliquot of this pooled sample was analyzed every fourteen samples in order to provide the measure of the system’s stability and performance. The system operation, data acquisition, and analysis were controlled and processed using Analyst 1.7.1,PeakView 2.2, and MultiQuant 3.0 softwares from AB Sciex Inc. (Vaughan, ON, Canada).

### 3.5. Determination of Total Phenolic Content (TPC) and Total Flavonoids Content (TFC)

TPC and TFC were measured on a UV-2550 Spectrophotometric Reader (Shimadzu Corporation, Kyoto, Japan). The absorbance was measured at 725 nm and 510 nm, respectively. All of the analyses were performed in triplicate. TPC and TFC analysis were performed using the photocolorimetric method, as described by Mohammed Moniruzzaman [41]. The TPC was expressed as milligrams of gallic acid equivalents per kilogram of honey (mg GAE/kg honey), and the standard curve was generated with gallic acid (10–160 μg. mL^−1^). The TFC were expressed as milligrams of epicatechin equivalents per kilogram of honey (mg EC/kg honey), and the standard curve was plotted using epicatechin (1–100 μg. mL^−1^).

### 3.6. Antioxidant Activity

Antioxidant activity assays including DPPH, ABTS and FRAP were studied as described by Habib et al. [42].

Radical scavenging activity assay (DPPH assay). The aqueous solution of honey (0.2 g. mL^−1^) was mixed with 3.8 mL of DPPH radical solution (0.25 mM). After incubating in the dark for 30 min, the absorbance of the solution was measured at 515 nm. The percentage of free radical scavenging activity that targeted DPPH was calculated using the following equation: DPPH radical savaging activity
(1)$$(\%\;\mathrm{inhibition})=\frac{A0-A1}{A0}\times 100$$
where *A*0 is the absorbance of the DPPH control, and *A*1 is the absorbance in the sample.

ABTS cation radical scavenging. The cation radical ABTS+ was synthesized by the reaction of a 7 mM ABTS solution with a 2.4 mM potassium persulfate solution. The mixture was kept at room temperature in the dark for 14 h. Afterwards, the ABTS+ solution was diluted with methanol until an absorbance of 0.73 ± 0.01 units at 734 nm was achieved. 1.0 mL of the honey sample (20% *w*/*v*) was mixed with 1.0 mL of fresh diluted ABTS solution. After incubation at room temperature for 7 min, the absorbance of the solution was measured to be 734 nm. The percentage inhibition calculated as ABTS radical scavenging activity was according to Equation (1), as provided above.

Ferric reducing/antioxidant power assay. The FRAP reagent was prepared before the test by mixing 100 mL of acetate buffer (300 mM, pH 3.6) with 10 mL of TPTZ solution (10 mM in 40 mM HCl) and 10 mL of ferric chloride (FeCl_3_, 20 mM). A total of 100 µL of the honey solution (0.2 g·mL^−1^) was mixed with 900 µL of ultrapure water, followed by adding 2.0 mL of the FRAP reagent. The mixture was then vortexed and incubated at 37 °C for 30 min. The absorbance was then determined to be 593 nm using ferrous sulfate standards (0, 0.1, 0.2, 0.5, 1, 1.5, 2.0 mM). The units used for the FRAP values was µmol of ferrous equivalents/100 g of honey sample.

### 3.7. Data Processing and Metabolomics Analysis

The first step of the metabolomics analysis was to collect information on the phenolic compounds in honey from the literature. Then, the extracted ion chromatogram (XIC) manager add-on in PeakView software 2.2 was used for isotope pattern matched peak mining of data files of honey samples. The parameters for the data mining experiments were as follows: RT window, 1–80 min; minimum intensity counts ≥100; S/N ratio ≥3; isotope pattern matching ≥80%. In addition to MS data, the spectra from MS/MS were also analyzed using the Fragments Pane add-on in PeakView software 2.2 to verify the fragmentation pattern of the detected compound and then matched with hits in the literature and the ChemSpider database (http://www.chemspider.com, accessed on 13 February 2023).

The peak areas of tentatively identified phenolic compounds in each honey sample were integrated using MultiQuant 3.0 software. The data set consisting of one hundred and eleven honey extracts from 37 honey samples was then subjected to PCA analysis using the R statistical package (Rx64 4.0.4). Pareto scaling of the data was performed to modify the weights of the respective variables. The validation of the obtained PCA model was performed by QC samples to ensure the performance of the models. In addition, the dataset was also subjected to orthogonal partial least squares-discriminant analysis (OPLS-DA) using SIMCA 14.1.

### 3.8. Statistical Analysis

The analyses were made in triplicate, and the results were expressed as the average ± standard deviation. Both the difference analysis and the correlation analysis were carried out with SPSS 20.0 software (Chicago, IL, USA).

## 4. Conclusions

The present study evaluated the antioxidant properties and phenolic profile of Shennongjia *A. cerana* honey in China. Furthermore, a total of 83 phenolic compounds were tentatively identified by HPLC-QTOF-MS/MS in this study. Among these compounds, the presence and levels of propolis-derived caffeic acid and pinobanksin ester derivatives in *A. cerana* honeys were lower than those in *A. mellifera* honeys. Moreover, thirteen compounds were tentatively identified as markers to distinguish between *A. cerana* and *A. mellifera* honey by PCA and OPLS-DA analysis. These compounds could be appropriate markers that should be studied further in the future.

## Figures and Tables

**Figure 1 molecules-28-03270-f001:**
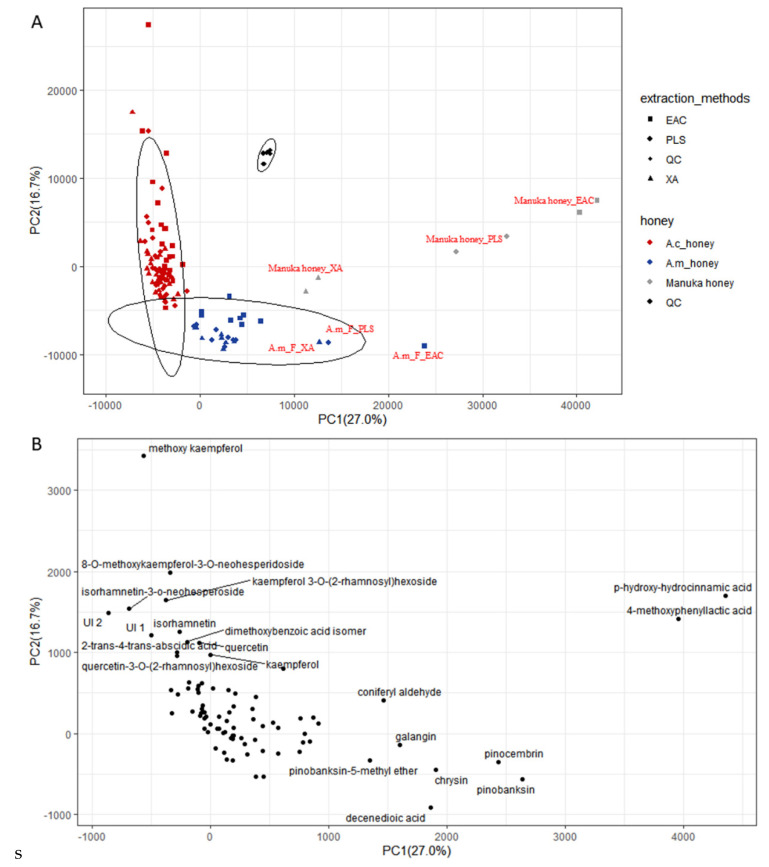
Principal components analysis (PCA) of *A. cerana* and *A. mellifera* honeys. Results of PCA of honeys: scores plot (**A**) and loadings plot (**B**).

**Figure 2 molecules-28-03270-f002:**
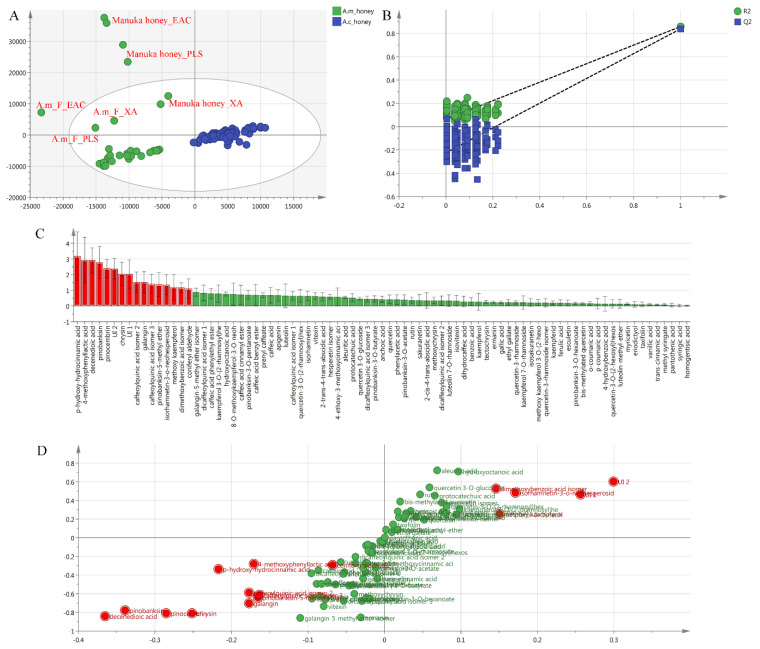
Orthogonal partial least squares-discriminant analysis (OPLS-DA) of *A. cerana* and *A. mellifera* honey. Score scattering plot of OPLS-DA (**A**) and corresponding validation plot (**B**); VIP (**C**) and S-plot (**D**) of OPLS-DA.

**Table 1 molecules-28-03270-t001:** Total phenolic and flavonoid contents and antioxidant activity of *A. cerana* and *A. mellifera* honeys.

	TPC (mg GAE/Kg)	TFC (mg EC/Kg)	FRAP (uM of Fe^2+^/100 g)	DPPH (%)	ABTS (%)
A.m_p1	104.33 ± 4.21	15.65 ± 1.59	60.13 ± 1.55	3.63 ± 0.21	55.95 ± 0.31
A.m_p2	239.12 ± 17.97	25.10 ± 0.86	175.13 ± 4.44	6.34 ± 0.19	68.78 ± 1.02
A.m_p3	225.38 ± 10.34	21.21 ± 0.51	199.50 ± 3.63	5.76 ± 0.14	75.95 ± 0.43
A.m_p4	177.04 ± 24.46	27.71 ± 1.53	93.25 ± 3.84	2.80 ± 0.11	58.78 ± 0.17
A.m_p5	186.88 ± 10.38	26.42 ± 1.48	133.25 ± 2.56	3.63 ± 0.16	65.68 ± 0.13
A.m_p6	158.29 ± 5.05	31.57 ± 1.04	100.13 ± 5.41	3.42 ± 0.08	57.84 ± 0.97
A.m_p7	130.48 ± 3.78	14.74 ± 0.71	78.25 ± 2.97	2.17 ± 0.24	54.73 ± 1.01
A.m_p8	172.91 ± 7.98	40.18 ± 1.02	111.38 ± 5.13	3.00 ± 0.35	60.27 ± 0.27
A.m_F	379.20 ± 25.86	42.76 ± 0.29	300.75 ± 4.32	8.93 ± 0.33	87.57 ± 0.34
MGO100+	622.16 ± 3.72	111.24 ± 3.91	719.50 ± 9.03	19.77 ± 0.56	94.73 ± 0.23
MGO250+	652.33 ± 9.70	110.42 ± 2.19	626.38 ± 6.81	16.02 ± 0.46	94.59 ± 0.13
A.c_1	550.70 ± 11.10	83.28 ± 0.35	607.63 ± 7.97	17.65 ± 0.56	92.97 ± 0.25
A.c_2	265.49 ± 5.57	45.8 ± 1.08	326.38 ± 4.56	9.18 ± 0.37	77.43 ± 0.48
A.c_3	450.61 ± 9.16	71.63 ± 0.29	375.75 ± 3.60	8.84 ± 0.18	88.11 ± 0.16
A.c_4	327.67 ± 3.79	49.65 ± 0.77	323.25 ± 4.63	7.72 ± 0.30	83.38 ± 0.30
A.c_5	271.57 ± 8.94	39.33 ± 0.86	330.13 ± 3.10	8.22 ± 0.10	82.16 ± 0.51
A.c_6	470.01 ± 12.73	102.24 ± 0.75	405.13 ± 5.78	11.60 ± 0.09	91.76 ± 0.30
A.c_7	340.00 ± 4.13	50.48 ± 2.16	423.88 ± 3.32	10.64 ± 0.32	91.49 ± 0.34
A.c_8	680.90 ± 35.80	87.51 ± 3.83	541.38 ± 5.66	13.60 ± 0.41	93.51 ± 0.42
A.c_9	334.67 ± 3.76	61.33 ± 1.65	344.50 ± 4.59	7.88 ± 0.43	84.31 ± 0.11
A.c_10	264.98 ± 4.79	45.78 ± 0.46	234.50 ± 3.13	7.38 ± 0.13	73.51 ± 0.32
A.c_11	270.66 ± 1.96	57.86 ± 1.24	245.75 ± 4.22	6.34 ± 0.09	71.76 ± 0.59
A.c_12	263.02 ± 2.23	44.05 ± 1.49	263.25 ± 3.50	8.22 ± 0.50	70.13 ± 0.46
A.c_13	382.51 ± 3.97	52.24 ± 0.75	382.63 ± 6.10	6.88 ± 0.20	84.05 ± 0.36
A.c_14	407.30 ± 17.02	59.14 ± 0.04	391.38 ± 5.66	9.30 ± 0.23	88.92 ± 0.62
A.c_15	327.02 ± 3.09	62.57 ± 0.59	321.38 ± 5.00	7.55 ± 0.48	76.89 ± 0.71
A.c_16	296.28 ± 8.15	60.85 ± 0.68	280.13 ± 2.94	7.38 ± 0.34	80.14 ± 0.58
A.c_17	326.52 ± 4.71	56.59 ± 2.14	387.63 ± 3.47	9.89 ± 0.35	87.84 ± 0.47
A.c_18	322.43 ± 7.27	43.62 ± 1.20	350.75 ± 3.66	6.88 ± 0.24	83.38 ± 0.52
A.c_19	329.70 ± 2.70	47.94 ± 0.41	383.88 ± 4.69	8.39 ± 0.33	84.05 ± 0.17
A.c_20	302.61 ± 6.95	48.88 ± 5.04	303.88 ± 3.17	7.38 ± 0.29	80.54 ± 0.63
A.c_21	360.04 ± 10.55	48.84 ± 4.56	360.75 ± 5.28	9.47 ± 0.53	74.86 ± 0.33
A.c_22	331.69 ± 12.15	35.87 ± 0.44	299.50 ± 2.80	8.01 ± 0.41	82.57 ± 0.46
A.c_23	343.23 ± 8.74	47.53 ± 1.88	335.13 ± 6.41	7.34 ± 0.12	83.24 ± 0.48
A.c_24	452.07 ± 9.32	58.75 ± 1.55	520.13 ± 4.04	11.39 ± 0.22	90.27 ± 0.40
A.c_25	317.91 ± 8.15	55.67 ± 0.83	362.00 ± 3.97	6.72 ± 0.34	74.46 ± 0.17
A.c_26	358.04 ± 2.44	42.76 ± 0.29	312.00 ± 3.67	6.26 ± 0.45	75.00 ± 0.60

Note: The TPC and TFC results are expressed as mean ± S.D. (*n* = 3).

**Table 2 molecules-28-03270-t002:** The average content of phenolic compounds (µg/100 g honey) in *A. cerana* and *A. mellifera* honeys.

		*A. mellifera* Honey (*n* = 9)	*A. cerana* Honey (*n* = 26)	Manuka Honey (*n* = 2)
Kaem	Kaem_XA	2.94 ± 2.31	2.25 ± 2.27 ^b^	0.51 ± 0.30
	Kaem_PLS	4.67 ± 3.36	3.56 ± 2.94 ^b^	0.99 ± 0.06
	Kaem_EAC	27.70 ± 12.59	47.72 ± 34.19 ^a^	9.92 ± 1.27
Quer	Quer_XA *	0.05 ± 0.10	0.85 ± 1.08 ^b^	0.19 ± 0.07
	Quer_PLS **	0.66 ± 0.33	2.20 ± 1.48 ^b^	0.99 ± 0.38
	Quer_EAC **	4.60 ± 1.65	14.85 ± 10.38 ^a^	5.88 ± 1.58
Pino	Pino_XA **	16.27 ± 25.18	0.37 ± 0.62 ^a^	22.68 ± 20.94
	Pino_PLS ***	15.53 ± 21.35	0.46 ± 0.81 ^a^	91.12 ± 1.07
	Pino_EAC ***	15.65 ± 18.29	0.60 ± 1.04 ^a^	46.73 ± 2.14
Gal	Gal_XA ***	2.11 ± 2.45	0.16 ± 0.26 ^b^	2.64 ± 2.44
	Gal_PLS ****	2.25 ± 1.82	0.22 ± 0.34 ^ab^	7.11 ± 0.35
	Gal_EAC ****	6.87 ± 5.90	0.55 ± 0.87 ^a^	14.58 ± 0.18
Ch	Ch_XA **	8.17 ± 11.92	0.17 ± 0.29 ^a^	34.89 ± 5.13
	Ch_PLS ****	5.65 ± 5.62	0.22 ± 0.41 ^a^	23.12 ± 2.73
	Ch_EAC ****	10.86 ± 10.32	0.48 ± 0.98 ^a^	17.52 ± 1.49
CTabsa	CTbasa_XA	56.67 ± 24.62	90.82 ± 60.22 ^a^	47.16 ± 17.02
	CTbasa_PLS	54.22 ± 21.24	83.60 ± 56.58 ^a^	65.78 ± 15.83
	CTbasa_EAC	54.05 ± 18.74	68.81 ± 45.63 ^a^	38.82 ± 15.09
4 Hba	4 Hba_XA	38.86 ± 26.60	29.22 ± 17.70 ^b^	10.77 ± 4.89
	4Hba_PLS *	105.21 ± 63.27	186.67 ± 88.91 ^a^	40.37± 13.53
	4Hba_EAC	113.86 ± 69.66	183.52 ± 98.30 ^a^	51.96 ± 7.39
Ru	Ru_XA **	1.41 ± 1.90	3.50 ± 1.61 ^a^	0
	Ru_PLS **	1.42 ± 1.53	3.84 ± 1.94 ^a^	0
	Ru_EAC **	0.39 ± 0.57	1.15 ± 0.67 ^b^	0
Tcina	Tcina_XA	3.79 ± 11.36	1.67 ± 4.30 ^ab^	0
	Tcina_PLS	2.43 ± 7.28	1.03 ± 3.99 ^b^	29.05 ± 4.37
	Tcina_EAC	10.23 ± 10.81	5.12 ± 7.16 ^a^	41.26 ± 14.91
Pcoa	Pcoa_XA *	8.34 ± 3.69	19.29 ± 13.13 ^a^	5.03 ± 0.20
	Pcoa_PLS *	13.09 ± 14.27	26.28 ± 16.11 ^a^	16.18 ± 2.15
	Pcoa_EAC*	12.30 ± 8.96	26.29 ± 17.60 ^a^	9.53 ± 2.64
Vana	Vana_XA ****	2.13 ± 2.06	0 ^c^	0
	Vana_PLS ****	7.56 ± 3.23	3.80 ± 1.53 ^b^	6.00 ± 0.55
	Vana_EAC	12.28 ± 10.25	10.18 ± 5.62 ^a^	11.38 ± 0.42
Cafa	Cafa_XA	49.18 ± 40.96	21.18 ± 37.83 ^a^	93.88± 8.32
	Cafa_PLS	50.45 ± 34.28	28.02 ± 64.41 ^a^	70.37 ± 7.29
	Cafa_EAC	51.93 ± 32.84	31.26 ± 54.89 ^a^	48.43± 7.62
Fera	Fera_XA	3.75 ± 1.92	2.99 ± 2.80 ^b^	0.43 ± 0.61
	Fera_PLS *	5.23 ± 4.84	2.71 ± 2.12 ^b^	2.05 ± 0.28
	Fera_EAC	9.21 ± 9.32	5.30 ± 3.25 ^a^	2.76 ± 0.52

“*” represents values that differed significantly between *A. cerana* and *A. mellifera* honeys for the same compound using the uniform extraction method, * means *p* < 0.05, ** means *p* < 0.01, *** means *p* < 0.001, **** means *p* < 0.0001. “^abc^” letters represent values that differed significantly among different extraction methods for the same compound.

**Table 3 molecules-28-03270-t003:** High resolution MS data and fragmentation of phenolic compounds identified in *A. cerana* and *A. mellifera* honeys.

No	RT (min)	Name	Formula	[M-H]^calculated^	[M-H]^experimental^	Error (ppm)	MS/MS	References	Detected in Honey Samples
		**Phenolic acids and abscidic acid**							
1	4.58	^c^ gallic acid	C7H6O5	169.0142	169.0142	−0.4	125	[27]	8/11 (*A. mellifera*), 26/26 (*A. cerana*)
2	8.18	^c^ protocatechuic acid	C7H6O4	153.0194	153.0193	0.3	109, 108	[28]	11/11 (*A. mellifera*), 25/26 (*A. cerana*)
3	9.65	^c^ homogentisic acid	C8H8O4	167.0350	167.0350	0.0	123, 93	[29]	5/11 (*A. mellifera*), 7/26 (*A. cerana*)
4	11.52	^c^ dihydrocaffeic acid	C9H10O4	181.0503	181.0506	−1.8	163, 135, 119, 93	[6]	8/11 (*A. mellifera*), 23/26 (*A. cerana*)
5	11.57	^a^ 4-hydroxybenzoic acid	C7H6O3	137.0244	137.0244	−0.1	93	std	All
6	12.36	^c^ caffeoylquinic acid isomer 1	C16H18O9	353.0874	353.0878	−1.1	191, 179, 135	[2]	11/11 *(A. mellifera*), 25/26 (*A. cerana*)
7	13.68	^c^ dimethoxybenzoic acid isomer	C9H10O4	181.0505	181.0506	−1.0	137, 121	[25]	9/11 (*A. mellifera*), 26/26 (*A. cerana*)
8	14.16	^b^ ethyl gallate	C9H10O5	197.0453	197.0455	−1.2	153, 109	chemspider	3/11 (*A. mellifera*), 22/26 (*A. cerana*)
9	14.73	^a^ benzoic acid	C7H6O2	121.0296	121.0295	1.0	108, 92	std	6/11 (*A. mellifera*), 20/26 (*A. cerana*)
10	17.08	^a^ vanillic acid	C8H8O4	167.0351	167.0350	1.0	152, 108, 123, 91	std	11/11 (*A. mellifera*), 25/26 (*A. cerana*)
11	17.64	^c^ esculetin	C9H6O4	177.0191	177.0193	−1.1	149, 133, 105, 89	[30]	All
12	17.93	^c^ phenylacetic acid	C8H8O2	135.0451	135.0452	−0.7	107	[27]	6/11 (*A. mellifera*), 22/26 (*A. cerana*)
13	17.97	^a^ caffeic acid	C9H8O4	179.0348	179.0350	−0.9	135	std	All
14	18.95	^c^ caffeoylquinic acid isomer 2	C16H18O9	353.0875	353.0878	−0.8	191, 179	[25]	11/11 (*A. mellifera*), 25/26 (*A. cerana*)
15	21.6	^c^ syringic acid	C9H10O5	197.0455	197.0455	−0.3	182, 166.9, 153, 138, 123, 95	[27]	9/11 (*A. mellifera*), 22/26 (*A. cerana*)
16	22.14	^b^ p-hydroxy-hydrocinnamic acid	C9H10O3	165.0556	165.0557	−1.0	147, 119, 103, 72.9	chemspider	All
17	23.98	^c^ caffeoylquinic acid isomer 3	C16H18O9	353.0880	353.0878	0.5	191, 179	[25]	9/11 (*A. mellifera*), 24/26 (*A. cerana*)
18	24.07	^a^ p-coumaric acid	C9H8O3	163.0399	163.0401	−1.3	119, 93	std	All
19	25.22	^c^ o-coumaric acid	C9H8O3	163.0397	163.0401	−2.0	119, 93	[27]	10/11 (*A. mellifera*), 23/26 (*A. cerana*)
20	26.43	^c^ methyl syringate	C10H12O5	211.0609	211.0612	−1.2	196, 181, 167, 153	[31]	4/11 (*A. mellifera*), 14/26 (*A. cerana*)
21	27.84	^c^ 4-methoxyphenyllactic acid	C10H12O4	195.0660	195.0663	−1.3	177, 134, 162, 149	[32]	8/11 (*A. mellifera*), 26/26 (*A. cerana*)
22	27.85	^c^ coniferyl aldehyde	C10H10O3	177.0552	177.0557	−2.6	162, 133, 117, 105	[33]	3/11 (*A. mellifera*), 26/26 (*A. cerana*)
23	28.88	^a^ ferulic acid	C10H10O4	193.0507	193.0506	0.4	178, 149, 134	std	All
24	31.6	^b^ 4-ethoxy-3-methoxycinnamic acid	C12H14O4	221.0818	221.0819	−0.7	193, 151, 179, 135	chemspider	2/11 (*A. mellifera*), 0/26 (*A. cerana*)
25	35.5	^c^ dicaffeoylquinic acid isomer 1	C25H24O12	515.1196	515.1195	0.2	353, 191, 179, 173	[28]	9/11 (*A. mellifera*), 23/26 (*A. cerana*)
26	36.61	^c^ 2-trans-4-trans-abscidic acid	C15H20O4	263.1282	263.1289	−2.5	219, 204, 201	[2]	All
27	37.87	^c^ dicaffeoylquinic acid isomer 2	C25H24O12	515.1197	515.1195	0.4	353, 191, 179, 173	[28]	9/11 (*A. mellifera*), 20/26 (*A. cerana*)
28	38.73	^a^ trans-cinnamic acid	C9H8O2	147.0451	147.0452	−0.6	119,103	std	7/11 (*A. mellifera*), 9/26 (*A. cerana*)
29	39.28	^a^ 2-cis-4-trans-abscidic acid	C15H20O4	263.1280	263.1289	−3.2	219, 204, 163, 152, 139	std	All
30	41.78	^c^ dicaffeoylquinic acid isomer 3	C25H24O12	515.1193	515.1195	−0.4	353, 191, 179, 173	[28]	9/11 (*A. mellifera*), 9/26 (*A. cerana*)
31	53.27	^c^ prenyl caffeate	C14H16O4	247.0973	247.0976	−1.3	179, 161, 135	[2]	9/11 (*A. mellifera*), 1/26 (*A. cerana*)
32	53.33	^c^ caffeic acid benzyl ester	C16H14O4	269.0817	269.0819	−1.0	178,161, 134	[2]	11/11 (*A. mellifera*), 2/26 (*A. cerana*)
33	55.35	^c^ caffeic acid phenylethyl ester	C17H16O4	283.0973	283.0976	−0.9	268, 215, 179, 161, 135	[2]	11/11 (*A. mellifera*), 4/26 (*A. cerana*)
34	57.83	^c^ caffeic acid cinnamyl ester	C18H16O4	295.0968	295.0976	−2.6	178, 134	[2]	11/11 (*A. mellifera*), 8/26 (*A. cerana*)
		**Flavonols**							
35	22.25	^c^ myricetin	C15H10O8	317.0302	317.0303	−0.3	299, 255, 206.9, 190.9, 163	[27]	1/11 (*A. mellifera*), 17/26 (*A. cerana*)
36	32.17	^c^ quercetin-3-O-(2-hexosyl) hexoside	C27H30O17	625.1415	625.1410	0.8	463, 300	[33]	9/11 (*A. mellifera*), 16/26 (*A. cerana*)
37	33.68	^c^ quercetin-3-O-(2-rhamnosyl)hexoside	C27H30O16	609.1469	609.1461	1.3	300	[34]	10/11 (*A. mellifera*), 24/26 (*A. cerana*)
38	33.74	^c^ methoxy kaempferol 3-O-(2-hexosyl) hexoside	C28H32O17	639.1575	639.1567	1.3	330, 314, 299	[33]	10/11 (*A. mellifera*), 21/26 (*A. cerana*)
39	34.77	^c^ 8-O-methoxykaempferol-3-O-neohesperidoside	C28H32O16	623.1638	623.1618	3.3	314, 315, 459, 608	[2]	9/11 (*A. mellifera*), 25/26 (*A. cerana*)
40	35.48	^c^ quercetin 3-O-glucoside	C21H20O12	463.0878	463.0882	−0.9	301, 300, 271	[33]	10/11 (*A. mellifera*), 26/26 (*A. cerana*)
41	35.57	^c^ kaempferol 3-O-(2-rhamnosyl)hexoside	C27H30O15	593.1523	593.1512	1.9	284	[33]	10/11 (*A. mellifera*), 26/26 (*A. cerana*)
42	35.91	^c^ isorhamnetin-3-o-neohesperoside	C28H32O16	623.1624	623.1618	1.0	314, 315, 459	[2]	11/11 (*A. mellifera*), 25/26 (*A. cerana*)
43	36.04	^a^ rutin	C27H30O16	609.1467	609.1461	1.0	300, 301	std	8/11 (*A. mellifera*), 26/26 (*A. cerana*)
44	38.17	^c^ quercetin-3-rhamnoside isomer	C21H20O11	447.0923	447.0933	−2.2	301, 300, 284, 255	[33]	10/11 (*A. mellifera*), 24/26 *(A. cerana*)
45	41.96	^c^ quercetin-3-rhamnoside	C21H20O11	447.0931	447.0933	−0.4	301, 300, 151	[2]	9/11 (*A. mellifera*), 23/26 (*A. cerana*)
46	43.25	^a^ quercetin	C15H10O7	301.0351	301.0354	−1.0	179, 151	std	All
47	46.92	^c^ kaempferol 7-O-rhamnoside	C21H20O10	431.0979	431.0984	−1.0	285, 257, 151	[33]	11/11 (*A. mellifera*), 24/26 (*A. cerana*)
48	47.71	^c^ methoxy kaempferol	C16H12O7	315.0506	315.0510	−1.3	300, 272, 255, 165.9	[33]	All
49	47.91	^a^ kaempferol	C15H10O6	285.0401	285.0405	−1.3	229, 185, 151, 239, 257	std	All
50	49.09	^c^ isorhamnetin	C16H12O7	315.0509	315.0510	−0.5	300, 151	[2]	11/11 (*A. mellifera*), 23/26 (*A. cerana*)
51	49.72	^c^ bis-methylated quercetin	C17H14O7	329.0666	329.0667	−0.3	314, 299, 271	[33]	All
52	53.14	^c^ kaempferid	C16H12O6	299.0554	299.0561	−2.5	284, 271, 255, 237, 211, 165	[33]	9/11 (*A. mellifera*), 5/26 (*A. cerana*)
53	55.6	^a^ galangin	C15H10O5	269.0451	269.0455	−1.6	213, 169	std	11/11 (*A. mellifera*), 13/26 (*A. cerana*)
54	56.32	^c^ galangin-5-methyl ether isomer	C16H12O5	283.0609	283.0612	−0.9	268, 239, 211	[2]	11/11 (*A. mellifera*), 16/26 (*A. cerana*)
		**Flavanonols**							
55	29.56	^c^ taxifolin	C15H12O7	303.0508	303.0510	−0.9	285, 275, 241, 177, 125	[35]	11/11 (*A. mellifera*), 25/26 (*A. cerana*)
56	40.72	^c^ pinobanksin-5-methyl ether	C16H14O5	285.0765	285.0768	−1.3	267, 252, 224, 165, 138	[2]	11/11 (*A. mellifera*), 7/26 (*A. cerana*)
57	41.99	^c^ pinobanksin	C15H12O5	271.0606	271.0612	−2.3	253, 197	[33]	All
58	53.72	^c^ pinobanksin-3-O-acetate	C17H14O6	313.0709	313.0718	−2.9	253, 271	[36]	9/11 (*A. mellifera*), 12/26 (*A. cerana*)
59	60.73	^c^ pinobanksin-3-O-butyrate	C19H18O6	341.1023	341.1031	−2.1	253, 271, 197	[2]	10/11 (*A. mellifera*), 4/26 (*A. cerana*)
60	65.35	^c^ pinobanksin-3-O-pentanoate	C20H20O6	355.1180	355.1187	−2.1	253, 271	[2]	11/11 (*A. mellifera*), 8/26 (*A. cerana*)
61	68.19	^c^ pinobanksin-3-O-hexanoate	C21H22O6	369.1333	369.1344	−2.8	300, 271, 253	[30]	10/11 (*A. mellifera*), 0/26 (*A. cerana*)
		**Flavanones**							
62	38.42	^c^ eriodictyol	C15H12O6	287.0552	287.0561	−3.3	151, 135	[33]	All
63	45.15	^c^ hesperetin isomer	C16H14O6	301.0714	301.0718	−1.1	164, 286	[29]	9/11 (*A. mellifera*), 26/26 (*A. cerana*)
64	52.06	^c^ isosakuranetin	C16H14O5	285.0766	285.0768	−1.0	165, 119	[14]	11/11 (*A. mellifera*), 19/26 (*A. cerana*)
65	52.47	^a^ pinocembrin	C15H12O4	255.0661	255.0663	−0.6	213, 171, 151	std	11/11 (*A. mellifera*), 16/26 (*A. cerana*)
66	52.57	^c^ sakuranetin	C16H14O5	285.0766	285.0768	−1.0	165, 119	[14]	11/11 (*A. mellifera*), 19/26 (*A. cerana*)
		**Flavones**							
67	34.09	^c^ isovitexin	C21H20O10	431.0982	431.0984	−0.3	385, 341, 311, 283, 251	[27]	2/11 (*A. mellifera*), 9/26 (*A. cerana*)
68	35.45	^b^ vitexin	C21H20O10	431.0980	431.0984	−0.9	341, 311, 283	chemspider	8/11 (*A. mellifera*), 10/26 (*A. cerana*)
69	41.25	^c^ luteolin 7-O-rhamnoside	C21H20O10	431.0977	431.0984	−1.5	285, 255, 227	[33]	10/11 (*A. mellifera*), 25/26 (*A. cerana*)
70	44.82	^c^ luteolin	C15H10O6	285.0400	285.0405	−1.5	133, 151, 175, 199	[27]	All
71	48.34	^c^ apigenin	C15H10O5	269.0454	269.0455	−0.5	225, 205, 151, 117	[27]	11/11 (*A. mellifera*), 25/26 (*A. cerana*)
72	49.48	^c^ luteolin-methyl-ether	C16H12O6	299.0559	299.0561	−0.7	284, 256, 190.9	[30]	All
73	50.29	^c^ tectochrysin	C16H12O4	267.0657	267.0663	−2.4	252, 224, 180	[33]	11/11 (*A. mellifera*), 6/26 (*A. cerana*)
74	54.32	^c^ methoxychrysin	C16H12O5	283.0605	283.0612	−2.4	268, 239, 211	[2]	7/11 (*A. mellifera*), 25/26 (*A. cerana*)
75	54.36	^a^ chrysin	C15H10O4	253.0504	253.0506	−1.0	209, 143	std	11/11 (*A. mellifera*), 17/26 (*A. cerana*)
76	56.86	^c^ ermanin	C17H14O6	313.0718	313.0718	0.2	298, 283, 255, 199	[30]	11/11 (*A. mellifera*), 2/26 (*A. cerana*)
		**Others**							
77	9.91	^c^ pantothenic acid	C9H17NO5	218.1028	218.1034	−2.7	146, 88, 71	[27]	All
78	16.25	^c^ UI 1	C10H7NO3	188.0352	188.0353	−0.5	144	[2]	11/11 (*A. mellifera*), 25/26 (*A. cerana*)
79	29.33	^c^ UI 2	C10H7NO3	188.0352	188.0353	−0.6	144	[2]	8/11 (*A. mellifera*), 26/26 (*A. cerana*)
80	37.35	^c^ anchoic acid	C9H16O4	187.0967	187.0976	−4.6	169, 125, 97	chemspider	All
81	39.01	^b^ hydroxyoctanoic acid	C8H16O3	159.1019	159.1027	−4.8	113	chemspider	All
82	41.04	^c^ decenedioic acid	C10H16O4	199.0970	199.0976	−2.9	155, 137, 181	[31]	All
83	59.03	^b^ aleuritic acid	C16H32O5	303.2172	303.2177	−1.7	285, 229	chemspider	All

^a^ confirmed with standard. ^b^ confirmed with chemspider. ^c^ confirmed with references.

## Data Availability

The data presented in this study are available in the article and Appendix A.

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
