# Peer review of "Evaluation of the Antioxidant Activities and Phenolic Profile of Shennongjia Apis cerana Honey through a Comparison with Apis mellifera Honey in China"

_molecules, 2023, doi:10.3390/molecules28073270_

Round 1

Reviewer 1 Report

This research is very extensive, detailed and relevant to the topic. However, it is only one of the many studies that have the same methodology, only samples of different types of honey (from different areas of the world).

The research is useful and new in terms of obtaining data on a specific type of honey limited to one area in the world.

Considering that a profile of phenolic compounds obtained by various extraction methods and metabolomics subjected to principal component analysis was done for the first time for a special type of honey, the paper can be accepted for publication because it can contribute to the collection of data on the characteristics of different types of honey.

Author Response

Thank you very much for your comments.

Shennongjia forestry district, the only well-preserved subtropical forest ecosystem in the middle latitudes of the world, located in Hubei Province in China. Being due to its superior climatic conditions and unique geographical environment, it has become a "refuge" for many rare plants and animals such as A. cerana. The traditional box farming technology and varietal purity of A. cerana in this region has high protection and research value. The pollen of abundant resources of bee plants and wild medicinal plants is a good source of honey, and the honey is the "shennong polyfloral honey" with local characteristics. In ancient China,“Shen Nong's Herbal Classic” have recorded the use of A. cerana honey from Shen-nong-jia district as the first choice of medicine. Therefore, it is necessary to evaluate the phenolic profile of this honey using different extraction methods and subject the data to metabolomics analysis.

Reviewer 2 Report

The article deals with a very specific local honey that is not commonly traded, on the other hand it contributes to knowledge, whether phenolic profile can be regarded as technique for studying the floral, geographical and honeybee origins of honeys.

The informative value of the article would be enhanced if also Apis mellifera honeys (which were used for comparison) originated in the same Shennongjia province and had a similar floral profile.

Thus, the study brings well processed data on phenolic content on A. cerana honey from certain locality, but not answering the question if some compounds can be real markers to distinguish between A. cerana and A. mellifera honeys in general.

In fact only A.m_p2 sample has at least partially similar floral profile comparing with A. cerana polyfloral honey. Some comparison honey samples can be even regarded as monofloral once (f.e. A.m_F, A.m_p3, A.m_p7, A.m_p8), which affects phenolic profiles.

In table 2 abbreviations used in the first column are not explained (even if it is clear what it means). 

Author Response

We thank the reviewer for critics, which greatly improved the manuscript.

The response to your comments is attached below. 

Reviewer 3 Report

The manuscript can be accepted in its current form

Author Response

Thanks a lot for your comments.

In ancient China,“Shen Nong's Herbal Classic” have recorded the use of A. cerana honey from Shen-nong-jia district as the first choice of medicine. Shen-nong-jia forestry district, the only well-preserved subtropical forest ecosystem in the middle latitudes of the world, located in Hubei Province in China. Being due to its superior climatic conditions and unique geographical environment, the pollen of abundant resources of bee plants and wild medicinal plants is a good source of honey, and the honey is the "shen-nong polyfloral honey" with local characteristics. In addition, Shennongjia Forest area is one of the protection bases of Apis cerana in China. The output of Apis cerana honey in this region is limited, some illegal vendors sell cheap Apis mellifera honey instead of Apis cerana honey, which seriously harms the interests of consumers and damages the credibility of Shennongjia honey as a geographical indication product. Therefore, it is necessary to evaluate the phenolic profile and the antioxidant properties of this kind of honey here.

Reviewer 4 Report

With this study the authors want to identify phenolic compounds as markers to distinguish A. cerana from A. mellifera honey. After many experimental tests they conclude that 13 compounds were identified. There are many inaccuracies and the text should be improved to make it easier to read. I found the manuscript difficult to understand.  I suggest only a few examples of inaccuracies, but the authors should improve the form of the entire manuscript.

-          Line 44. Add references after the statement

-          Line 44-45 and line 48-51. You repeat the same concept twice

-          2.1 Chemicals. I do not think is necessary to specify the origin of the chemical compounds. The chemical characteristics should be the same regardless of the origin

-          Line 97. The results should be mentioned in the appropriate paragraph, not in Materials and methods

-          Table 1. Specify somewhere (e.g. in line 204 where A. mellifera and A. cerana is written) what the abbreviation "A.m_p1 etc" and "A.c._1 etc" means

-          Are lines 244-245 part of the Table 2 legend? In the present form the lines seems to be part of the text

Author Response

(The authors gave the same response as above.)
